# COVID-19 Vaccine Acceptance among Health Care Workers in the United States

**DOI:** 10.3390/vaccines9020119

**Published:** 2021-02-03

**Authors:** Rahul Shekhar, Abu Baker Sheikh, Shubhra Upadhyay, Mriganka Singh, Saket Kottewar, Hamza Mir, Eileen Barrett, Suman Pal

**Affiliations:** 1Department of Internal Medicine, University of New Mexico Health Sciences Center, Albuquerque, NM 87106, USA; rshekhar@salud.unm.edu (R.S.); supadhyay@salud.unm.edu (S.U.); ebarrett@salud.unm.edu (E.B.); spal@salud.unm.edu (S.P.); 2Department of Medicine, University Hospitals, Case Western Reserve University, Cleveland, OH 44139, USA; mriganka.singh@uhhospitals.org; 3Department of Medicine, Division of Hospital Medicine, University of Texas Health San Antonio, San Antonio, TX 78229, USA; kottewar@uthscsa.edu; 4Data Analyst, University of New Mexico Health Sciences Center, Albuquerque, NM 87106, USA; hmir@salud.unm.edu

**Keywords:** Covid-19, vaccine, healthcare workers, United States

## Abstract

Background: Acceptance of the COVID-19 vaccine will play a major role in combating the pandemic. Healthcare workers (HCWs) are among the first group to receive vaccination, so it is important to consider their attitudes about COVID-19 vaccination to better address barriers to widespread vaccination acceptance. Methods: We conducted a cross sectional study to assess the attitude of HCWs toward COVID-19 vaccination. Data were collected between 7 October and 9 November 2020. We received 4080 responses out of which 3479 were complete responses and were included in the final analysis. Results: 36% of respondents were willing to take the vaccine as soon as it became available while 56% were not sure or would wait to review more data. Only 8% of HCWs do not plan to get vaccine. Vaccine acceptance increased with increasing age, education, and income level. A smaller percentage of female (31%), Black (19%), Lantinx (30%), and rural (26%) HCWs were willing to take the vaccine as soon as it became available than the overall study population. Direct medical care providers had higher vaccine acceptance (49%). Safety (69%), effectiveness (69%), and speed of development/approval (74%) were noted as the most common concerns regarding COVID-19 vaccination in our survey.

## 1. Introduction

COVID-19 has rapidly become a major public health crisis, affecting 86.4 million individuals, and causing 1.9 million deaths globally by January of 2021. The US has reported more than 21 million cases and 357,000 deaths as of 5 January 2021 [1]. To curb this pandemic, apart from effective public health measures such as social distancing, wearing face masks, hand washing, and avoidance of crowded indoor spaces, educating the general population, efficacious vaccination is emerging as essential to mitigating disease and death [2,3,4,5,6].

Uptake of any COVID-19 vaccine is an important challenge to address. In a recent survey, more than one-third of lay respondents were unsure or did not intend to take the vaccine [7]. Clinicians are an important source of information for vaccines and physician communication can improve adherence to vaccination recommendations [8,9,10]. Thus, the role of healthcare workers (HCWs) becomes particularly important in advising patients and communities, and as well as through role modeling behavior. HCWs are prioritized among the high-risk groups who are considered as candidates for early vaccination. As such, it is important to consider HCW attitudes about COVID-19 vaccination to better address barriers to widespread vaccination.

## 2. Methods

Design: We conducted a cross sectional study to assess the attitude of HCWs toward COVID-19 vaccination. An online English questionnaire was created using REDCap electronic data capture tools hosted at the University of New Mexico. The survey was modified from a previously published general population survey to capture more information pertinent to healthcare workers. Data were collected anonymously, and no personally identifying information was collected. This study was approved by the University of New Mexico Hospitals Institutional Review Board (study ID 20-515).

Sampling: A snowball sampling was utilized. The survey tool was distributed via links posted on social media platforms in various HCW groups and distributed to administrative leaders at five major hospital systems—two in the state of New Mexico, and one each in the states of Texas, Missouri, and Ohio, to disseminate among their employees. Data were collected between 7 October and 9 November 2020.

Participants: All adults (>18 years of age) living in the US and working in a healthcare setting regardless of patient care contact and role in health care settings were eligible to participate in the study. Informed consent was obtained prior to enrollment in the study. Incomplete responses were excluded from the analysis.

## 3. Measures

Demographic information collected included age, gender, ethnicity, race, state of primary residence in majority of last six months, occupation, marital status, the number of household members excluding participant, annual household income, location of healthcare setting (rural, suburban or urban), education level and political orientation (Conservative/Republican, unaffiliated, Democrat/Liberal, or do not wish to answer). Self-perceived risk of COVID-19 was gauged by the question “Do you think you are at risk of getting COVID-19 in the next 1 year?” The responses allowed for graded self-perceived risk (“No I am confident I won’t get infected”; “Yes I am concerned that I will get mild symptoms which will probably not require hospitalization”; “Yes I am concerned that I will get moderate symptoms which will probably need hospitalization”; “Yes I am concerned that I will get severe symptom which will probably require admission to the intensive care unit”; “I believe I already have the disease and I am immune to it (not diagnosed by a test)”; “No, I already have recovered and won’t get re-infected (diagnosed by a test)”.

Exposure to COVID-19 was assessed by the questions “Have you directly or indirectly taken care of the COVID-19 patients?” and “Have you, your family member or someone you know been diagnosed with COVID-19 (excluding your patients)?”

Acceptance of COVID-19 vaccine was assessed by the question “When COVID-19 vaccination becomes available, would you take it?” Participants could choose responses from among the options: “Yes, as soon as I can get it”; “Yes, only if it is required by employer”; “No, I will wait for 3 months to review safety profile”; “No, I will wait for 6 months to review safety profile”; “No, I will wait for at least 1 year to review safety profile”; “I will not get the vaccine”; “Not sure”.

Attitude toward the vaccination was assessed by agreement with perception/concern statements as measured on a Likert scale [11]. A five point Likert scale with options from Strongly disagree to strongly agree was used and respondents were instructed to select the option that best aligned with their views. General perception about vaccines were assessed by statements “I do not believe vaccines work”; “I do not believe vaccines are safe”; “I do not get vaccinated for religious reasons”; “I do not get vaccinated for reasons of personal freedom/choice”; “I do not get vaccinated for a fear of needles/doctors/hospitals”. Concerns regarding COVID-19 vaccine were evaluated by statements ”I am worried about the safety/adverse effects of COVID-19 vaccine”; “I am worried about effectiveness of COVID-19 vaccine”; “I am worried about the out of pocket cost/Insurance coverage of the vaccine”; “I am concerned about adverse effect of vaccine on my pre-existing conditions”; “I am worried about the rapidity of the development and approval of COVID-19 vaccine”; “I do not need the vaccine for my risk level”; “I am worried about the rapidity of the development and approval of COVID-19 vaccine”. Agreement was measured on a Likert scale from strongly disagree to strongly agree.

## 4. Outcome

We divided HCWs into four major categories: Direct medical provider (DMP) which includes physician/resident/medical student/advanced practice provider (including Nurse Practitioner and physician’s assistant); direct patient care provider (DPCP) including registered nurse/patient care technician/paramedic/rehabilitation services (respiratory, physical, occupational, or speech therapist), nutritionist, social worker, case manager, care coordinator; administrative staff, and others without direct patient care. If the participant is DMP, their primary specialty was also recorded and is divided into primary medical, primary surgical, diagnostics, and others.

The primary outcome of the survey was whether HCWs are willing to take the COVID-19 vaccine or not. Responses were collected against willingness to “take it as soon as it becomes available” and “yes but only if it is required by employer” (grouped into one as yes responses), “wait for safety data review for 3 months”, “wait for safety data review for 6 months” or “wait for safety data review for a year” or “not sure” (grouped into one response “wait for review”), and “not willing” to take it.

We performed a multinomial logistic regression because of the overall description of the data. We split the sample into three groups according to the primary outcome variable, would a participant take the COVID-19 vaccination immediately, or would wait to review safety data, or would not take the vaccination at all. To analyze the association between vaccine acceptance and participant characteristics, we used likelihood-ratio test leading to the derived chi-square and *p*-values. Statistical significance was assumed for *p* < 0.05. We used the R programming language (R Foundation for Statistical Computing, Vienna, Austria. http://www.R-project.org/) to develop the model and analyze its results.

## 5. Results

3479 HCWs completed the survey. Most participants were younger than 40 years of age (1877, 54%), female (2598, 75%), White (2803, 81%), completed a bachelor’s degree or higher (2788, 80%) identified as Democrat or Liberal (1521, 44%), and had no chronic medical conditions (2039, 59%). Most reported working in an urban area (2229, 64%), in primary medical and medical subspecialties (54%) and provided direct patient care (79%). The majority of participants perceived themselves to be at risk for acquiring COVID-19 (3043, 87%) and 21% think they will acquire serious disease requiring admission to hospital (747) but less than eight percent participant were confident that they would not get the disease (267, 8%). Roughly half of the HCWs have directly taken care of COVID-19 positive patients (1570, 45%). Most of the participants believed that COVID-19 vaccination should be voluntary (1665, 48%). Additional sample characteristics are shown in Table 1.

Only about one-third (1247, 36%) of the respondents were willing to take a COVID-19 vaccine as soon as it became available at the time of the survey. A majority of the HCW were not sure or would wait to review safety data before getting vaccinated (1953, 56%). Among the respondents who want to wait, 11% will like to wait for 3 months, 10% will like to wait for 6 months, and 20% will like to wait at least 1 year. Only 8% (279) of respondents were unwilling to take the vaccine at all.

A significant association was noted between the choice that participants make about receiving COVID-19 vaccination and multiple predictor variables (Table 2). We note that acceptance of COVID-19 vaccination increased with increasing age. In the 18–30 age group only 34% of respondents were willing to take COVID-19 vaccine as soon as it became available which increased to 47% in the >70 age group. A similar trend was noted with education and income level; increasing education and income levels represent a higher proportion of HCWs willing to take the vaccine as soon as it becomes available.

Differences in vaccine acceptance were noted by gender and racial identity. Female HCWs had lower vaccine acceptance at 31% compared to male HCWs (49%) and trans/non-binary HCWs (43%). Black HCWs had lower acceptance (19%) with the majority choosing to wait to review safety data (65%) whereas Asian HCWs had high vaccine acceptance (44%). A majority of Native American HCWs (80%) and all Native Hawaiian/Other pacific islander HCWs (100%) chose to wait to review data for COVID vaccine. Vaccine acceptance was lower among those identifying as Hispanic or Latino (30%). Geographical variation in vaccine acceptance was also noted in our survey with West having the lowest (33%) and South having the highest (48%) vaccine acceptance. HCWs employed in rural settings had lower acceptance (26%) of the vaccine. Those identifying as a Democrat/Liberal had higher vaccine acceptance (42%). HCWs who believe themselves to be immune to COVID-19 and those who feel confident they will not get infected had the highest rates of refusing COVID-19 vaccination at 22% and 27% respectively. HCWs who had not taken care of COVID-19 patients had higher rates of vaccine refusal (9.2%).

Vaccine acceptance varied by occupational role in healthcare. DMPs had higher vaccine acceptance (49%) than administrative staff (34%) and others without direct patient care (33%). DPCPs had the lowest vaccine acceptance (27%) with nearly two-thirds (62%) of DPCP choosing to wait to review safety data.

Acceptance of COVID-19 vaccine was also associated with a plan to recommend vaccination to friends and family, and with a higher likelihood of wanting COVID-19 vaccine for HCWs to be mandatory. Of note, a large majority of respondents who plan not to get a COVID-19 vaccine would also not recommend the vaccine to friends or family and want vaccination to be voluntary.

While overall concerns regarding vaccination in general were low, concerns regarding COVID-19 vaccines were prevalent (Figure 1). Most HCWs believe that in general vaccination works (90%), is safe (86%) and did not mention personal (87%) or religious belief (95%) as a reason for not vaccinating. Most participants endorsed concerns (agree or strongly agree) about vaccine safety/adverse effects (69%), effectiveness (69%), and rapidity of development/approval (74%). A majority of HCWs trust their doctors and healthcare professionals recommending the COVID-19 vaccine (73%) but nearly half of the respondents do not trust information provided by the government about COVID-19 and its severity (46%) and one-third do not trust regulatory authorities like CDC or FDA overseeing the vaccine development and safety (34%).

## 6. Discussion

Since the announcement of efforts to develop a COVID-19 vaccine, several surveys have been conducted to gauge public perception and acceptance of the vaccine through 2020 [7,12,13]. Most surveys have focused on the general population. However, the rollout of the vaccine is tiered to various subgroups of the population based on limited availability, and HCWs are among the first subgroups of the US population to have access to the vaccine. HCWs are also likely to be an important source of information about the vaccine for the general population. As such, it is crucial to assess predictors of vaccine acceptance among HCWs which will help institutions and policy-makers target resources to maximize the uptake. To the best of our knowledge, this is one of the first and largest surveys of HCWs in the US about COVID-19 vaccination.

In our survey, only one-third of respondents were amenable to COVID-19 vaccination immediately, while more than half of respondents preferred to defer their decision until reviewing more data. This contrasts with the findings of a general population survey in April 2020 that reported half the participants were willing to take the vaccine. Another study in May 2020 reported 67% acceptance of COVID-19 vaccination among the US adults [7,12]. The high percentage of respondents waiting to review more data in our study is expected as HCWs are more likely to base healthcare decisions on published scientific literature of efficacy and safety which was underway during the time of our survey. This also highlights the importance of publication and dissemination of scientific data regarding the vaccine which will be a crucial factor to determine eventual uptake of the vaccine among HCWs.

Overall, only 8% respondents said they would refuse the COVID-19 vaccine which shows a potential high uptake of the vaccine among HCWs. Increasing vaccination acceptance has substantial benefits. To eventually slow down the spread of COVID-19 and its mortality, it is imperative to achieve herd immunity by vaccination before immunity by natural infection. With an estimated reproductive number(R) of 3, the threshold herd immunity for COVID-19 will be achieved by immunizing at least 70% of the population assuming the vaccine is 100% effective [14]. With recent Emergency Use Authorizations by the Food and Drug Administration of 2 vaccines in the US reported to have almost 95% effectiveness in phase III clinical trials, this number could be even higher. Thus, it is imperative to vaccinate a maximum number of HCWs to prevent the infection among HCWs and loss of critical workforce. Studies have shown that vaccination of HCWs with influenza vaccine decreases patient mortality and staff absenteeism [15,16,17,18]. It would be reasonable to expect a similar benefit with COVID-19 vaccination. It is important to note that 97% of HCWs in our survey had received an influenza vaccine in the previous year indicating a generally favorable perception of vaccination.

The low initial acceptance of COVID-19 vaccine among HCWs could also have broader consequences. Studies have shown that HCWs who are vaccinated are more likely to recommend vaccines to friends, family, and their patients [19,20,21]. This has also been borne out in our study where we see a strong association among HCWs who plan to be vaccinated and plan to recommend the vaccine to friends and family.

COVID-19 vaccine acceptance increased with increasing age, income, and education level. This mirrors the trends seen with the general population surveys conducted in the US [7,12]. Higher vaccine acceptance with increasing age could be due to higher perceived vulnerability, as also suggested by Detoc et al. [22]. Black race also had lower vaccine acceptance while Asian race had higher vaccine acceptance. Vaccine acceptance was also lower in HCWs identifying as Hispanic or Latino. Racial differences in vaccination for influenza has been previously reported [23,24]. Fisher et al. also reported similar findings in their general population survey for COVID-19 vaccine uptake [7]. This mirrors the trends seen with general population surveys [7,12]. The toll of COVID-19 pandemic has disproportionately affected low-income communities and Black, Indigenous, and people of color [25]. Lower vaccine uptake could exacerbate the health inequities among these communities. Targeted messaging and outreach would be required to achieve higher vaccination rates [26]. It will also be important to understand and address the factors driving low vaccine acceptance which would be instrumental in addressing the key concerns and directing the resources to increase the uptake.

HCWs identifying as Conservative or Republican had lower acceptance of the vaccine whereas those identifying as Liberal or Democrat had higher vaccine acceptance. This is in line with other general US population surveys [7,13,27]. This difference could be in part due to differential messaging regarding vaccines in news and social media targeting these populations and in part due to very divergent responses to the pandemic by political leaders of major parties [28]. A unified and consistent messaging from political leaders will be essential to bridge this gap.

Vaccine acceptance was higher among HCWs involved in direct patient care and in HCWs with chronic medical conditions. This could mirror the trends in perceived risk of COVID-19 infection. More HCWs who perceived themselves to be at risk for COVID-19 infection were willing to accept than refuse the vaccine. However, even among HCWs directly involved in patient care, vaccine acceptance was also lower among HCWs identified as DPCP than among DMP. Less favorable attitude toward the influenza vaccine among nurses as compared to physicians has previously been reported [29]. A recent survey out of Hong Kong reported low COVID-19 vaccine acceptance among nurses as well [30]. This trend is concerning since DPCP (such as nurses, respiratory therapists, etc.,) often have more direct and prolonged patient contact. They are, therefore, at high risk of infection. DPCP are also one of the key resources of the healthcare system where critical shortages have been noted during the pandemic. They therefore represent a key subgroup whose health is essential to continue the care of patients with COVID-19 and understanding and addressing their concerns will be crucial.

To understand the factors driving vaccine uptake, we assessed HCWs’ attitude toward vaccination and toward COVID-19 vaccine. Concerns regarding vaccination in general were low in our study, consistent with other studies that show generally positive attitudes of healthcare workers toward vaccination. However, concerns specific to COVID-19 vaccination were prevalent. We found frequent concerns regarding the vaccine efficacy, adverse effects, and rapidity of development (Figure 1). This was particularly noted among HCWs who do not plan to take the COVID-19 vaccine. Of note, while HCWs who did not want to be vaccinated reported poor trust in regulatory authorities and government, their trust in medical professionals prescribing the vaccine was somewhat higher. This could suggest an important role for dissemination of information through medical agencies and professional societies to increase the uptake among HCWs.

A major strength of our study is the large sample of HCWs surveyed. Our survey population is also diverse with representation from different genders, age groups, ethnic and racial backgrounds, and roles in healthcare.

We recognize the limitations of our study. Because of the sampling method being snowball sampling, selection bias would limit the generalizability of our findings since our study population may not be representative of all US HCWs. Despite these limitations, these findings are not inconsistent with the findings from previous studies about HCW vaccine hesitancy. The survey questionnaire was available in English and distributed in an online format, which can further introduce selection bias favoring English-literate HCWs and those with access to the Internet. Social desirability bias may also affect the interpretation of our study results, though the responses were anonymized to minimize this. Most importantly, our study was conducted when information regarding COVID-19 vaccines under development were limited and findings of the clinical trials had not been made public. As such, it is possible that with this information now being publicly available, both vaccine acceptance and attitude toward COVID-19 vaccines have changed.

## 7. Conclusions

Immediate acceptance of a COVID-19 vaccine is low, with the majority of HCWs choosing to wait to review more data before deciding on personal vaccination. However, a very small percentage of respondents plan to refuse vaccination, suggesting the potential for high uptake. Overall attitudes toward vaccination were positive but specific concerns regarding COVID-19 vaccine are prevalent. Differences in vaccine acceptance were noted along demographic lines in our subjects with lower acceptance in historically under-served communities. Addressing barriers to vaccination among these groups will be essential to avoid exacerbating health inequities laid bare by this pandemic.

## Figures and Tables

**Figure 1 vaccines-09-00119-f001:**
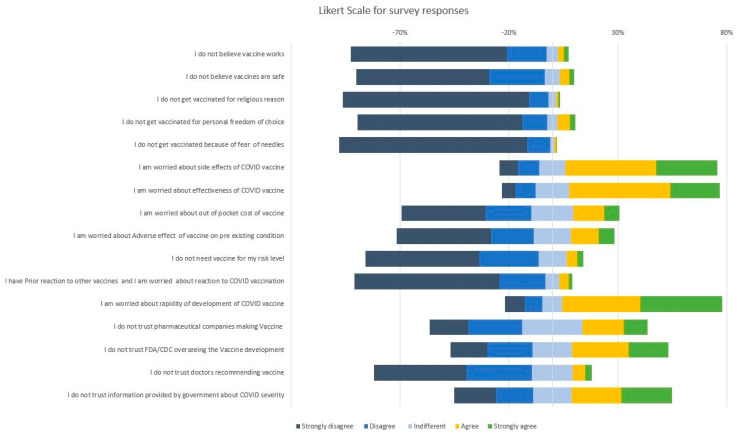
Attitude of healthcare workers toward vaccines.

**Table 1 vaccines-09-00119-t001:** Participant characteristics.

Variable	N = 3479 ^1^
**Age**	
18–30 years	816 (23%)
31–40 years	1061 (30%)
41–50 years	686 (20%)
51–60 years	571 (16%)
61–70 years	326 (9.4%)
>70 years	19 (0.5%)
**Gender**	
Female	2598 (75%)
Male	864 (25%)
Trans/ Gender non-binary/not specified above	7 (0.2%)
Do not wish to reply	10 (0.3%)
**Ethnicity**	
Hispanic or Latino	560 (16%)
NOT Hispanic or Latino	2763 (79%)
Unknown/Not Reported	36 (1.0%)
Do not wish to answer	120 (3.4%)
**Race**	
White or Caucasian	2803 (81%)
Asian	218 (6.3%)
Black or African American	74 (2.1%)
Native Americans/Alaska Native	30 (0.9%)
Native Hawaiian or Other Pacific Islander	6 (0.2%)
More Than One Race	126 (3.6%)
Unknown/Other	70 (2.0%)
Do not wish to answer	152 (4.4%)
**State of Residence**	
Midwest	1433 (41%)
North East	81 (2.3%)
South	314 (9.0%)
West	1651 (47%)
**Occupation**	
Direct Patient Care Providers (DPCP)	1573 (45%)
Direct Medical Providers (DMP)	1207 (35%)
Administrative staff working in hospital without direct patient contact	295 (8.5%)
Others without direct patient contact	404 (12%)
**Primary Area of Work**	
Primary medical and medical subspecialty	1882 (54%)
Primary surgical and surgical subspecialty	363 (10%)
Diagnostic subspecialty	246 (7.1%)
Others	988 (28%)
**Annual Income**	
<$30,000	117 (3.4%)
$30,001–$70,000	741 (21%)
$70,001–$100,00	759 (22%)
$100,001–$150,000	796 (23%)
>$150,001	1066 (31%)
**Health Care facility in**	
Rural area	293 (8.4%)
Suburban area	957 (28%)
Urban area	2229 (64%)
**Education**	
No formal education	1 (<0.1%)
High school graduate, diploma or the equivalent (for example: GED)	46 (1.3%)
Some college credit, no degree	169 (4.9%)
Trade/technical/vocational training	111 (3.2%)
Associate degree	364 (10%)
Bachelor’s degree	1046 (30%)
Master’s degree	606 (17%)
Professional degree	297 (8.5%)
Doctorate degree	839 (24%)
**Political Identification**	
Conservative-Republican	746 (21%)
Democrat-Liberal	1519 (44%)
Unaffiliated	640 (18%)
Do not wish to answer	574 (16%)
**Medical comorbidities**	
No Chronic Condition	2039 (59%)
Diabetes Mellitus Type-1/Type-2	136 (3.9%)
Heart Disease	36 (1.0%)
Hypertension	449 (13%)
COPD Asthma—Lung Disease	318 (9.1%)
Obesity BMI > 30	461 (13%)
Cancer	125 (3.6%)
Immuno-compromised/on immunosuppressants	110 (3.2%)
Smoking	288 (8.3%)
Other Medical Conditions	505 (15%)
**Have you, your family member or someone you know been diagnosed with COVID-19 (Excluding your patients)**	
I was Diagnosed with COVID-19	90 (2.6%)
Family Member was Diagnosed with COVID-19	447 (13%)
**Someone I personally know was Diagnosed with COVID-19**	1793 (52%)
**No one I personally know was Diagnosed with COVID-19**	1400 (40%)
**Do you think you are at risk of getting COVID-19 in the next 1 year?**	
I believe I already have the disease and I am immune to it (Not diagnosed by a test)	138 (4.0%)
No, I am confident I won’t get infected	266 (7.6%)
No, I already have recovered and won’t get re-infected (Diagnosed by a test)	34 (1.0%)
Yes, I am concerned that I will get mild symptoms which will probably not require hospitalization	2294 (66%)
Yes, I am concerned that I will get moderate symptoms which will probably need hospitalization	572 (16%)
Yes, I am concerned that I will get severe symptom which will probably require admission to the Intensive care unit	175 (5.0%)
**Have you directly or indirectly taken care of the COVID-19 patients?**	
No	1263 (36%)
Yes, but no direct patient contact	646 (19%)
Yes, I have direct patient contact	1570 (45%)
**Would you take the COVID-19 Vaccine**	
No	279 (8.0%)
Wait for Review	1953 (56%)
Yes	1247 (36%)
**Would you advise friends and family to get vaccinated for COVID-19?**	
No	519 (15%)
Not sure	1376 (40%)
Yes	1584 (46%)
**COVID-19 Vaccine for health care workers should be:**	
Mandated by the employer, like Influenza vaccine	792 (23%)
Mandated by the Federal government for all health care workers	338 (9.7%)
Mandated by the State government for all health care workers	99 (2.8%)
Not sure	585 (17%)
Voluntary	1665 (48%)

^1^ Statistics presented: n (%). Direct Patient Care Provider: Registered Nurse/Patient care technician/Paramedics/Rehab services—Respiratory therapist/Physical therapist/Occupation therapist/Speech/Nutritionist/social workers/care coordinators. Direct Medical Provider: Physician/Resident/Medical students and Advanced practice provider (Nurse Practitioner/Physician’s assistant. Diagnostic Subspeciality: Radiology/Cath/Endoscopy/Pulmonary/Echo/Sleep Laboratory Technicians.

**Table 2 vaccines-09-00119-t002:** Participant characteristics by vaccine acceptance.

Variable	No, *n* = 279 ^1^	Wait for Review, *n* = 1953 ^1^	Yes, *n* = 1247 ^1^	*p*-Value ^2^
**Age**				<0.001 *
18–30 years	72 (8.8%)	467 (57%)	277 (34%)	
31–40 years	83 (7.8%)	615 (58%)	363 (34%)	
41–50 years	70 (10%)	389 (57%)	227 (33%)	
51–60 years	43 (7.5%)	306 (54%)	222 (39%)	
61–70 years	10 (3.1%)	167 (51%)	149 (46%)	
>70 years	1 (5.3%)	9 (47%)	9 (47%)	
**Gender**				<0.001 *
Female	240 (9.2%)	1540 (59%)	818 (31%)	
Male	37 (4.3%)	402 (47%)	425 (49%)	
Trans/Gender non-binary/not specified above	0 (0%)	4 (57%)	3 (43%)	
Do not wish to reply	2 (20%)	7 (70%)	1 (10%)	
**Ethnicity**				<0.001 *
Hispanic or Latino	55 (9.8%)	337 (60%)	168 (30%)	
NOT Hispanic or Latino	191 (6.9%)	1536 (56%)	1036 (37%)	
Unknown/Not Reported	8 (22%)	16 (44%)	12 (33%)	
Do not wish to answer	25 (21%)	64 (53%)	31 (26%)	
**Race**				<0.001 *
White or Caucasian	221 (7.9%)	1545 (55%)	1037 (37%)	
Asian	1 (0.5%)	122 (56%)	95 (44%)	
Black or African American	12 (16%)	48 (65%)	14 (19%)	
Native Americans/Alaska Native	3 (10%)	24 (80%)	3 (10%)	
Native Hawaiian or Other Pacific Islander	0 (0%)	6 (100%)	0 (0%)	
More Than One Race	4 (3.2%)	82 (65%)	40 (32%)	
Unknown/Other	9 (13%)	41 (59%)	20 (29%)	
Do not wish to answer	29 (19%)	85 (56%)	38 (25%)	
**State of Residence**				0.003 *
Midwest	146 (10%)	776 (54%)	511 (36%)	
North East	2 (2.5%)	45 (56%)	34 (42%)	
South	11 (3.5%)	151 (48%)	152 (48%)	
West	120 (7.3%)	981 (59%)	550 (33%)	
**Occupation**				<0.001 *
Direct patient care providers (DPCP)	187 (12%)	969 (62%)	417 (27%)	
Direct Medical Provider (DMP)	30 (2.5%)	582 (48%)	595 (49%)	
Administrative staff working in hospital without direct patient contact	25 (8.5%)	170 (58%)	100 (34%)	
Others without direct patient contact	37 (9.2%)	232 (57%)	135 (33%)	
**Annual Income**				<0.001 *
<$30,000	19 (16%)	53 (45%)	45 (38%)	
$30,001–$70,000	81 (11%)	420 (57%)	240 (32%)	
$70,001–$100,00	80 (11%)	463 (61%)	216 (28%)	
$100,001–$150,000	60 (7.5%)	472 (59%)	264 (33%)	
>$150,001	39 (3.7%)	545 (51%)	482 (45%)	
**Health Care facility in**				<0.001 *
Rural area	52 (18%)	164 (56%)	77 (26%)	
Suburban area	98 (10%)	523 (55%)	336 (35%)	
Urban area	129 (5.8%)	1266 (57%)	834 (37%)	
**Education**				<0.001 *
No formal education	0 (0%)	1 (100%)	0 (0%)	
High school graduate, diploma, or the equivalent (for example: GED)	9 (20%)	26 (57%)	11 (24%)	
Some college credit, no degree	24 (14%)	94 (56%)	51 (30%)	
Trade/technical/vocational training	18 (16%)	69 (62%)	24 (22%)	
Associate degree	56 (15%)	220 (60%)	88 (24%)	
Bachelor’s degree	109 (10%)	627 (60%)	310 (30%)	
Master’s degree	40 (6.6%)	362 (60%)	204 (34%)	
Professional degree	6 (2.0%)	139 (47%)	152 (51%)	
Doctorate degree	17 (2.0%)	415 (49%)	407 (49%)	
**Political Identification**				<0.001 *
Conservative-Republican	93 (12%)	390 (52%)	263 (35%)	
Democrat-Liberal	42 (2.8%)	833 (55%)	644 (42%)	
Unaffiliated	48 (7.5%)	373 (58%)	219 (34%)	
Do not wish to answer	96 (17%)	357 (62%)	121 (21%)	
**Did you get the Influenza vaccine Last Year?**				<0.001 *
No	50 (43%)	56 (48%)	10 (8.6%)	
Yes	229 (6.8%)	1897 (56%)	1237 (37%)	
**If you have Children under 18 years old, Have they been vaccinated for other diseases**				<0.001 *
No	38 (26%)	67 (46%)	41 (28%)	
Not Applicable	120 (6.0%)	1132 (56%)	754 (38%)	
Yes	121 (9.1%)	754 (57%)	452 (34%)	
**Medical Condition = Chronic Condition**				0.046
No	107 (7.4%)	819 (57%)	514 (36%)	
Yes	172 (8.4%)	1134 (56%)	733 (36%)	
**Diabetes Mellitus Type-1/Type-2**				0.076
No	272 (8.1%)	1876 (56%)	1195 (36%)	
Yes	7 (5.1%)	77 (57%)	52 (38%)	
**Heart Disease**				0.067
No	276 (8.0%)	1935 (56%)	1232 (36%)	
Yes	3 (8.3%)	18 (50%)	15 (42%)	
**Hypertension**				0.8
No	253 (8.3%)	1701 (56%)	1076 (36%)	
Yes	26 (5.8%)	252 (56%)	171 (38%)	
**COPD Asthma—Lung Disease**				0.701
No	258 (8.2%)	1778 (56%)	1125 (36%)	
Yes	21 (6.6%)	175 (55%)	122 (38%)	
**Obesity BMI>30**				0.022 *
No	250 (8.3%)	1671 (55%)	1097 (36%)	
Yes	29 (6.3%)	282 (61%)	150 (33%)	
**Cancer**				0.399
No	274 (8.2%)	1882 (56%)	1198 (36%)	
Yes	5 (4.0%)	71 (57%)	49 (39%)	
**Immuno-compromised/on immunosuppressants**				0.354
No	269 (8.0%)	1889 (56%)	1211 (36%)	
Yes	10 (9.1%)	64 (58%)	36 (33%)	
**Smoking**				0.013 *
No	262 (8.2%)	1778 (56%)	1151 (36%)	
Yes	17 (5.9%)	175 (61%)	96 (33%)	
**Other Medical Conditions**				0.629
No	227 (7.6%)	1663 (56%)	1084 (36%)	
Yes	52 (10%)	290 (57%)	163 (32%)	
**I was Diagnosed with COVID-19**				0.009 *
No	272 (8.0%)	1901 (56%)	1216 (36%)	
Yes	7 (7.8%)	52 (58%)	31 (34%)	
**Family Member was Diagnosed with COVID-19**				0.695
No	244 (8.0%)	1712 (56%)	1076 (35%)	
Yes	35 (7.8%)	241 (54%)	171 (38%)	
**Someone I personally know was Diagnosed with COVID-19**				0.003 *
No	168 (10.0%)	924 (55%)	594 (35%)	
Yes	111 (6.2%)	1029 (57%)	653 (36%)	
**No one I personally know was Diagnosed with COVID-19**				0.143
No	134 (6.4%)	1181 (57%)	764 (37%)	
Yes	145 (10%)	772 (55%)	483 (34%)	
**Do you think you are at risk of getting COVID-19 in next 1 year?**				<0.001 *
I believe I already have the disease and I am immune to it (Not diagnosed by a test)	30 (22%)	68 (49%)	40 (29%)	
No, I am confident I won’t get infected	71 (27%)	128 (48%)	67 (25%)	
No, I already have recovered and won’t get re-infected (Diagnosed by a test)	4 (12%)	18 (53%)	12 (35%)	
Yes, I am concerned that I will get mild symptoms which will probably not require hospitalization	153 (6.7%)	1283 (56%)	858 (37%)	
Yes, I am concerned that I will get moderate symptoms which will probably need hospitalization	15 (2.6%)	350 (61%)	207 (36%)	
Yes, I am concerned that I will get severe symptom which will probably require admission to the Intensive care unit	6 (3.4%)	106 (61%)	63 (36%)	
**Have you directly or indirectly taken care of the COVID-19 patients?**				<0.001 *
No	116 (9.2%)	714 (57%)	433 (34%)	
Yes, but no direct patient contact	45 (7.0%)	358 (55%)	243 (38%)	
Yes, I have direct patient contact	118 (7.5%)	881 (56%)	571 (36%)	
**I would get the vaccine to prevent COVID-19 in myself**				<0.001 *
No	277 (20%)	816 (58%)	324 (23%)	
Yes	2 (<0.1%)	1137 (55%)	923 (45%)	
**I would get the vaccine to prevent COVID-19 in friends and family members**				<0.001 *
No	276 (21%)	723 (56%)	286 (22%)	
Yes	3 (0.1%)	1230 (56%)	961 (44%)	
**I would get the vaccine to prevent COVID-19 in community**				<0.001 *
No	277 (26%)	560 (52%)	237 (22%)	
Yes	2 (<0.1%)	1393 (58%)	1010 (42%)	
**I would not get the vaccine**				<0.001 *
No	3 (<0.1%)	1787 (59%)	1235 (41%)	
Yes	276 (61%)	166 (37%)	12 (2.6%)	
**Would you advise friends and family to get vaccinated for COVID-19?**				<0.001 *
No	230 (44%)	272 (52%)	17 (3.3%)	
Not sure	49 (3.6%)	1162 (84%)	165 (12%)	
Yes	0 (0%)	519 (33%)	1065 (67%)	
**COVID-19 Vaccine for health care workers should be:**				<0.001 *
Mandated by the employer, like Influenza vaccine	0 (0%)	245 (31%)	547 (69%)	
Mandated by the Federal government for all health care workers	0 (0%)	90 (27%)	248 (73%)	
Mandated by the State government for all health care workers	0 (0%)	33 (33%)	66 (67%)	
Not sure	5 (0.9%)	457 (78%)	123 (21%)	
Voluntary	274 (16%)	1128 (68%)	263 (16%)	

^1^ Statistics presented: n (%); ^2^ statistical tests performed: likelihood-ratio test, chi-square test; * statistical significance accepted for *p* < 0.05; direct medical provider: physician/resident/medical students and advanced practice provider (nurse practitioner/physician’s assistant); diagnostic subspeciality: radiology/cath/endoscopy/pulmonary/echo/sleep laboratory technicians.

## Data Availability

The data presented in this study are available on request from the corresponding author. The data are not publicly available due to participant’s privacy concerns.

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
