# Peer review of "COVID-19 Vaccine Acceptance among Health Care Workers in the United States"

_vaccines, 2021, doi:10.3390/vaccines9020119_

Round 1

Reviewer 1 Report

I read with interest the article on vaccine-acceptance among health care workers in the US. Important and timely research. Please find my detailed comments below, mainly concerning clarifications. One major major limitation is the clear selection bias since the selected group does not seem representative for the HCW in the US. This issue should be addressed in more depth. It is however challenging to get such a high number of participants, and there will always be a certain degree of selection bias. I guess you don’t have figures on how people found out about this study?

  • Abstract: % per “lower acceptance group” is not clear. What does it mean? 10% of the Black (not) wanting vaccination? 10% of all not willing vaccine being black?
  • I would stress in abstract that only 8% would not take the vaccine.
  • Introduction: maybe good to mention date for the estimate of cases and deaths since the cumulative numbers are still going up
  • Please clarify if numbers are for US or elsewhere (if relevant) - unfortunately the pandemic is everywhere, and vaccine acceptance isn’t only a problem in the US
  • Did snowball sampling contribute to the relative low acceptance? Same circles may think alike? A-priori I would think that those participate are more willing, or maybe less willing…
  • Questionnaire was probably only in English? Could this have contributed to selection bias? It’s not only the educated personal working in the hospitals who are invited… but 80% were highly educated – wow, and I guess their income is not representative for the income all over the US with 75% earning >70,000 USD/year (or is that family income?!). 24% with a PhD is also indicating a major selection bias here – I doubt that this is representative for all HCW…
  • Participant eligibility can be described more clearly: so they all have to live in this region and be employed by one of these 5 hospitals? Or also volunteers?
  • Just wondering if this question about politics would have been included if there weren’t all those troubles around the presidential elections and the controversial messages from the US leadership during this pandemic. Very good it was included and that you thought about it!
  • Table 1: I am a bit confused about the categorisations by race and ethnicity… aren’t they basically used to describe the same thing?
  • A few typos throughout

Author Response

Reviewer 1

Dear Respectful Reviewer of the Journal of Vaccines,

Thank you for the reviewers’ valuable feedback and comments. Below you will find our answers to the reviewer questions. In the revised Manuscript file, the changes are highlighted in blue color. 

 Reviewer Comment - One major limitation is the clear selection bias since the selected group does not seem representative of the HCW in the US. This issue should be addressed in more depth. It is however challenging to get such a high number of participants, and there will always be a certain degree of selection bias. I guess you don’t have figures on how people found out about this study?

Response – We agree with the reviewer that due to the method of snowball sampling, selection bias is a significant limitation of our study. We have edited the discussion to reflect this and it now reads

Line 282 – We recognize the limitations of our study. Due to the sampling method being snowball sampling, selection bias would limit the generalizability of our findings since our study population may not be representative of all US HCWs. Despite these limitations, these findings are not inconsistent with findings from previous studies about HCW vaccine hesitancy.

Since the same survey link was shared across different platforms – social media such as Twitter, Facebook, and institutional emails of five academic centers – we are unable to determine the percentage of responses from different sources. Moreover, since the respondents were encouraged to share the survey link with their peers, it further limits our ability to identify how respondents found out about the study.

  • Abstract: % per “lower acceptance group” is not clear. What does it mean? 10% of the Black (not) wanting vaccination? 10% of all not willing vaccine being black?

Response – We have edited the abstract for clarity.

Line 23 – A smaller percentage of female (31%), Black (19%), Latinx (30%), and rural (26%) HCWs were willing to take the vaccine as soon as it became available than the overall study population.

  • I would stress in the abstract that only 8% would not take the vaccine.

Response – We have edited the abstract to include the suggestion.

Line 21 - Only 8% of HCWs do not plan to get vaccinated.

  • Introduction: maybe good to mention date for the estimate of cases and deaths since the cumulative numbers are still going up

Response – Date for the estimates has been added.

Line 33 - as of January 5,2021[1]

  • Please clarify if numbers are for US or elsewhere (if relevant) - unfortunately the pandemic is everywhere, and vaccine acceptance isn’t only a problem in the US

Response – In the introduction section, we have clarified which numbers reflect the global burden of disease and which represent the US. The sentence now reads as  -

Line 31- COVID-19 has rapidly become a major public health crisis, affecting 86.4 million individuals, and causing 1.9 million deaths globally. The US has reported more than 21 million cases and 357,000 deaths as of January 5,2021[1]

  • Did snowball sampling contribute to the relative low acceptance? Same circles may think alike? A-priori I would think that those participate are more willing, or maybe less willing…

Response – We cannot rule out the effect of snowball sampling on the vaccine acceptance in our population however, as reviewer notes, due to social desirability bias/voluntary response bias, we would have expected a higher vaccine acceptance among voluntary respondents. Overall high vaccine acceptance - albeit delayed - suggests against like-mindedness as a significant factor.

  • Questionnaire was probably only in English? Could this have contributed to selection bias? It’s not only the educated personal working in the hospitals who are invited… but 80% were highly educated – wow, and I guess their income is not representative for the income all over the US with 75% earning >70,000 USD/year (or is that family income?!). 24% with a PhD is also indicating a major selection bias here – I doubt that this is representative for all HCW

Response – We agree with the reviewer and have added this as a limitation to our study.

Line 286 - The survey questionnaire was available in English and distributed in an online format, which can further introduce selection bias favoring English-literate HCWs and those with access to the internet.

  • Participant eligibility can be described more clearly: so they all have to live in this region and be employed by one of these 5 hospitals? Or also volunteers?

Response – Participants were adults (>18years of age), living in the US, and working in a healthcare setting. Participation was not limited to the 5 hospital systems and our respondents were from all geographic regions of the US – Midwest, Northeast, South and West - as noted in table 1.  To better define the eligibility, we have edited the methods section –

Line 60 - All adults (>18 years of age) living in the US and working in a healthcare setting regardless of patient care contact and role in health care settings were eligible to participate in the study.

  • Just wondering if this question about politics would have been included if there weren’t all those troubles around the presidential elections and the controversial messages from the US leadership during this pandemic. Very good it was included and that you thought about it!

Response – We thank the reviewer for their appreciation of our effort to include political identity as a demographic factor. Indeed, it was the political discourse around vaccines and around public health measures like mask wearing that encouraged us to consider this addition as a recognition that public health does not operate in isolation from polity.

  • Table 1: I am a bit confused about the categorisations by race and ethnicity… aren’t they basically used to describe the same thing?

Response – We separated Latinx ethnic identity from other racial identities for two reasons -
1. Latinx identity – referring to ancestry in Latin America - is not mutually exclusive to racial identities of White, Black, Native American, etc.

  1. Latinx identity is a recognized underserved population in US healthcare and has suffered a higher burden of COVID-19, therefore we felt it imperative to identify trends in this vulnerable population to mitigate potential for disparities.

  • A few typos throughout

Response - We have scrutinized the manuscript and corrected for errors in spelling, grammar, and syntax.

Reviewer 2 Report

This manuscript by Rahul Shekhar et al. demonstrate very interesting data on COVID19 vaccine acceptance among HCWs.

This manuscript deserve some revisions before acceptance for publication.

Methods : How many distribution of the survey tool link were made?

Results :

Table 1 : fuse the entire three last lists in a unique one.

Table 2 :

Why have only Hispanic or Latino ethnicity been identified?

Have results been stratified by HCWs categories? It could be interesting to see if results varies between nurse and MD for example... please complete.

Could the authors highlight (when 3 or more categories exist to answer a question) the category significantly different ? Moreover, was multiple testing been taken into account and corrected ?

Discussion :

Use a unique reference presentation for the whole manuscript.

Could the authors compare the results obtained for COVID19 vaccine to HCW's adherence to other vaccine (as Influenza, see Web-based analysis of adherence to influenza vaccination among French healthcare workers Pichon et al. JClinVirol 2019,doi:org/10.1016/j.jcv.2019.04.008 )

Author Response

REVIEWER 2:

Dear Respectful Reviewer of the Journal of Vaccines,

Thank you for the reviewers’ valuable feedback and comments. Below you will find our answers to the reviewer questions. In the revised Manuscript file, the changes are highlighted in blue color. 

Methods: How many distribution of the survey tool link were made?

Response – We used a single survey tool link which was initially distributed to points of contact in five institutions to then be distributed among all their employees via institutional emails; and was also posted on two social media platforms – Twitter and Facebook (on private and public healthcare worker groups) – through accounts of the authors. Participants were further encouraged to share the link with their peers in keeping the snowball sampling design. Therefore, we are unable to determine how many distributions were eventually made during the study duration.

Results :

Table 1: fuse the entire three last lists in a unique one.

Response – We are unclear on what the reviewer refers to as “three last lists” and would welcome further clarification on this to help improve Table 1.

Table 2 :

Why have only Hispanic or Latino ethnicity been identified?

Response – Hispanic or Latino ethnicity was specifically identified since in the general US population survey by Fisher et al (from which our survey has been adapted), Hispanic/Latinx ethnicity was associated with a lower vaccine acceptance.

Have results been stratified by HCWs categories? It could be interesting to see if results varies between nurse and MD for example... please complete.

Response – HCWs have been categorized as - Direct medical providers (DMP) which includes physicians, trainee physicians, and advanced practice providers; Direct patient care providers (DPCP) which includes RNs, paramedical and therapy staff; and others without direct patient care.  We have noted the differences in vaccine acceptance among these groups and reported it in the following text –

Line 168 - DMPs had higher vaccine acceptance (49%) than administrative staff (34%) and others without direct patient care (33%). DPCPs had the lowest vaccine acceptance (27%) with nearly two-thirds (62%) of DPCP choosing to wait to review safety data.

We also discuss that the lower acceptance among DPCP such as RNs is worrisome due to more direct / prolonged contact with patients.

Line 257 - However, even among HCWs directly involved in patient care, vaccine acceptance was also lower among HCWs identified as DPCP than among DMP. This is concerning since DPCP (such as nurses, respiratory therapists, etc.) often have more direct and prolonged patient contact. They are, therefore, at high risk of infection. DPCP are also one of the key resources of the healthcare system where critical shortages have been noted during the pandemic. They therefore represent a key subgroup whose health is essential to continue the care of patients with COVID-19 and understanding and addressing their concerns will be crucial.

Could the authors highlight (when 3 or more categories exist to answer a question) the category significantly different? Moreover, was multiple testing been taken into account and corrected?

Response – We highlight the difference in primary outcome variable (vaccine acceptance) among the categories in each predictor variable in table 2. Significant difference is accepted for p<0.05. This clarification has been added to the legend for Table 2.

Discussion :

Use a unique reference presentation for the whole manuscript.

Response – References have been edited to have a homogenous format.

Could the authors compare the results obtained for COVID19 vaccine to HCW's adherence to other vaccine (as Influenza, see Web-based analysis of adherence to influenza vaccination among French healthcare workers Pichon et al. JClinVirol 2019,doi:org/10.1016/j.jcv.2019.04.008 )

Response – In our discussion, we have noted that 97% of our survey population reported receiving an influenza vaccine in the previous year, in sharp contrast to the lower acceptance of COVID-19 vaccine.

Line 224

 - It is important to note that 97% of HCWs in our survey had received an influenza vaccine in the previous year indicating a generally favorable perception of vaccination.

We note similarities in the findings of Pichon et al regarding Influenza vaccine to our findings regarding COVID-19 vaccination. However, we are hesitant to compare these two studies since the information regarding the influenza vaccine – efficacy, safety, adverse effect profile – were readily available at the time of study by Pichon et al whereas very limited data regarding COVID-19 vaccines was publicly available at time of our study.

Reviewer 3 Report

Reviewer’s comments and suggestions

The study focused on the acceptance of COVID-19 vaccines among health care workers in the United States. The authors conducted a cross sectional study to monitor the attitude of healthcare workers on vaccination. The study collected the data between October 7th and November 9th, 2020. They have included 3479 responses and were included in the final analysis.

The result noted that 36% of respondents were taking a chance of the vaccine while 56% were not sure and needed time so that more databases can form. Vaccine acceptance increased with increasing age, education, and income level. Lower acceptance was noted in females (31%), Black (10%), Latinx (30%) and Conservative/Republican (21%) health care workers, and those working in a rural setting (26%). In this survey, the important parameter was that the direct medical care providers had higher vaccine acceptance (49%). Moreover, safety (69%), effectiveness (69%), and approval rate (74%) were renowned as the most common concerns concerning vaccination. Please incorporate the below suggestions. 

Decision: Minor revision

  1. The author's number should be corrected it’s not sequential 
  2. Line 29-30, Its increasing need to update with the time (either you need to mention the recent information)
  3. Line 50-51, Need a number?
  4.  Line 52, Need to write about the merit of sampling in consent to your study design
  5. Line 54, You need to mention the five major hospital systems, I also have a concern about writing the US as the data represents a small group of patients so it would be better to write the region because your results do not represent the whole US population
  6. Line 83-84, is it valid in that way? I mean the author has to put some previous study references
  7. Line 94, Explain this scale as well, needed to discuss very minutes things in your paper. It's really important while observing the epidemiological data.
  8. Check the typo error line 98 and 102
  9. Line 120, Better to present a ray diagram to show study design including the recruited person, also point out exclusion and inclusion criteria if any
  10. In my view No need to include the political identification
  11. I think the authors have to categorize the data of normal and comorbidity patients and see the differences among them.
  12. Line 142-143, What would be the possible reason for this
  13. The word utilized in the sentence needs to rewrite “Someone Personal was Diagnosed with COVID-19”
  14. Line 183-184, Please include some of them and the most important one is that the author should write the novelty of his study in the first para.
  15. Line 197, What would be the probable reason for this, need to discuss other studies
  16. Line 209, with R of 3
  17. Line 224-225 Discuss your results, I found that the authors did not want to discuss their results completely. 
  18. Line 230-231 put previous study references
  19. Line 245, why this was? Line 256, which table? Please mention the information in the text
  20. Line 277-279 Not understand the lines why the author put here
  21. Please check the reference number 12, 19 and 22.

Author Response

Dear Respectful Reviewer,

Round 2

Reviewer 1 Report

No further comments

Reviewer 3 Report

The authors have addressed all concerns.